

# Jellyfish extract induces apoptotic cell death through the p38 pathway and cell cycle arrest in chronic myelogenous leukemia K562 cells

Sun-Hyung Ha[1,*], Fansi Jin[2,*], Choong-Hwan Kwak[1], Fukushi Abekura[1], Jun-Young Park[1], Nam Gyu Park[3], Young-Chae Chang[4], Young-Choon Lee[5], Tae-Wook Chung[6], Ki-Tae Ha[6], Jong-Keun Son[2], Hyeun Wook Chang[2] and Cheorl-Ho Kim[1]

[1] Molecular and Cellular Glycobiology Unit, Department of Biological Sciences, Sungkyunkwan University, Suwon City, Kyunggi-Do, Republic of Korea

[2] College of Pharmacy, Yeungnam University, Gyeongsan, Gyeongsang, Republic of Korea

[3] Department of Biotechnology, College of Fisheries Sciences, Pukyung National University, Busan, Republic of Korea

[4] Research Institute of Biomedical Engineering and Department of Medicine, Catholic University of Daegu, Daegu, Republic of Korea

[5] Department of Medicinal Biotechnology, College of Health Science, Dong-A University, Busan, Republic of Korea

[6] Division of Applied Medicine, School of Korean Medicine, Pusan National University, Yangsan City, Gyeongsangnam-Do, Republic of Korea

[*] These authors contributed equally to this work.

Corresponding authors
Hyeun Wook Chang, hwchang@yu.ac.kr
Cheorl-Ho Kim, chkimbio@skku.edu

## ABSTRACT

Jellyfish species are widely distributed in the world's oceans, and their population is rapidly increasing. Jellyfish extracts have several biological functions, such as cytotoxic, anti-microbial, and antioxidant activities in cells and organisms. However, the anti-cancer effect of Jellyfish extract has not yet been examined. We used chronic myelogenous leukemia K562 cells to evaluate the mechanisms of anti-cancer activity of hexane extracts from Nomura's jellyfish in vitro. In this study, jellyfish are subjected to hexane extraction, and the extract is shown to have an anticancer effect on chronic myelogenous leukemia K562 cells. Interestingly, the present results show that jellyfish hexane extract (Jellyfish-HE) induces apoptosis in a dose- and time-dependent manner. To identify the mechanism(s) underlying Jellyfish-HE-induced apoptosis in K562 cells, we examined the effects of Jellyfish-HE on activation of caspase and mitogen-activated protein kinases (MAPKs), which are responsible for cell cycle progression. Induction of apoptosis by Jellyfish-HE occurred through the activation of caspases-3,-8 and -9 and phosphorylation of p38. Jellyfish-HE-induced apoptosis was blocked by a caspase inhibitor, Z-VAD. Moreover, during apoptosis in K562 cells, p38 MAPK was inhibited by pretreatment with SB203580, an inhibitor of p38. SB203580 blocked jellyfish-HE-induced apoptosis. Additionally, Jellyfish-HE markedly arrests the cell cycle in the G0/G1 phase. Therefore, taken together, the results imply that the anti-cancer activity of Jellyfish-HE may be mediated apoptosis by induction of caspases and activation of MAPK, especially phosphorylation of p38, and cell cycle arrest at the Go/G1 phase in K562 cells.

## INTRODUCTION

Jellyfish belong to the phylum Cnidaria; they are lower animals with a non-polyp form. They consist of an umbrella-typed bell and trailing tentacles and are made of gelatin-based compounds. Jellyfish are widely distributed in the world's oceans and are rapidly increasing in population (*Lucas Brotz et al., 2012*). Evolutionarily, jellyfish can be traced in the seas going back approximately 0.7 billion years (*Cartwright et al., 2007*); they are the oldest known multi-organ animal (*Angier, 2011*).

Stinging jellyfish is known to have poisonous venom such as cardiovascular and pore-forming toxins (*Li et al., 2014*; *Burnett, 2009*; *Yanagihara & Shohet, 2012*; *Tibballs et al., 2011*; *Marino et al., 2007*; *Mariottini, 2014*; *Jennifer purcell, Shin-ichi Uye & Wen-Tseng Lo, 2007*). The venom is mainly located in extracts from the tentacles, which are made up of nematocysts and other cell types. Recently, jellyfish have been regarded as a beneficial resource having tumor-cytotoxic (*Kang et al., 2009*), anti-microbial (*Ovchinnikova et al., 2006*) and anti-oxidative (*Yu et al., 2005*) properties. To date, various kinds of jellyfish venoms have been reported to have diverse potential effects in the health science field as novel bioactive therapeutic agents with water-soluble or lipid-soluble compounds (*Leone et al., 2013*; *Rocha et al., 2011*; *Haefner, 2003*; *Schwartsmann et al., 2003*; *Mariottini & Pane, 2013*; *Morabito et al., 2015*; *Yanagihara & Shohet, 2012*; *Tibballs et al., 2011*). For example, a mucin glycoprotein of Nemopilema nomurai is a putative drug candidate (*Masuda et al., 2007*; *Ohta et al., 2009*). Several pharmacological properties including angiotensin-I-converting enzyme inhibitory (*Zhuang, Sun & Li, 2010*), anti-hypertensive (*Zhuang et al., 2012*), and immune-stimulatory activities (*Morishige et al., 2011*) have been reported. Green fluorescent protein from the jellyfish Aequorea victoria (*Shimomura, 1979*) is also a well-known biomarker used in the life sciences.

Chronic myeloid leukemia (CML) is a myeloproliferative tumor, which grows from a malignant myeloid lineage. Philadelphia chromosome translocation between chromosomes 9 and 22 is known to be a causative factor in CML with tyrosine kinase activity (*Jabbour & Kantarjian, 2014*). Although CML is treated with gleevec, imatinib mesylate (STI-571) as a tyrosine kinase inhibitor, STI-571-resistant patients have appeared, requiring other drug options. To overcome the drug resistance problem, many studies on CML-targeting drugs have been done by various researchers using natural products and cell-derived compounds (*Kwak et al., 2015*; *Jin et al., 2014*; *Kang et al., 2008*; *Jin et al., 2008b*; *Motomura et al., 2008*; *Jin et al., 2008a*).

Induction of apoptosis, or programmed cell death, is a preferred strategy for bringing about CML regression. Apoptosis is a biological adaptation that maintains homeostasis. Two main apoptotic pathways, extrinsic and intrinsic apoptosis, are known. The extrinsic apoptotic pathway is caused by death receptors (DRs). Apoptosis is induced by DRs that are related to activation of caspase-8. The other pathway, the intrinsic pathway, involves the mitochondria (*Ola, Nawaz & Ahsan, 2011*). Most chemotherapeutic drugs
act via stimulating apoptosis of cancer cells. However, toxicity and resistance lead to failure of chemotherapy in CML patients (*Ghasemian et al., 2015*). For that reason, natural compounds are increasingly considered alternative treatment that has potentials for therapy. The cell cycle is intimately involved with cell proliferation and survival of human cancer cells. In normal cells, the cell cycle regulates cellular division and replication, whereas in cancer cells, cell cycle regulation fails, leading to uncontrolled cell proliferation. Therefore, as an alternative anti-cancer strategy, cyclin dependent kinase (CDK) and cyclin, cell cycle regulators have been considered for patients with CML. Because cells are arrested at cell cycle checkpoints in order to repair cellular damage and control cell cycle-related genes, cell cycle-related therapy is a promising strategy for cancer treatment (*Senderowicz, 2002*).

We have searched for potential therapeutic agents with effects against CML based on natural compounds, especially from marine sources. In this study, we carried out activity-based pharmacological assays using extracts from Nomura's jellyfish obtained through solvent-based fractionation, and several anti-cancer compounds were obtained from fractionation using extraction with different solvents. Then, we demonstrated that jellyfish hexane extract has potential anti-cancer activity in K562 cells, as treatment of cells induces apoptosis and cell cycle arrest.

## MATERIALS AND METHODS

### Extraction of Jellyfish hexane extract (Jellyfish-HE)

*Stomolophus nomurai* (Nomura's jellyfish) were harvested from the shore near Busan, Korea. The voucher specimen has been deposited after classical identification in the invertebrate animals stocks of College of Fisheries Sciences, Pukyung National University, Busan, Korea (Prof NG Park). In order to dry the raw materials, the jellyfish has been harvested from coastal fishery and the water content was naturally removed using a house sieve. Then, the roughly dried jellyfish (100 g) was vacuum-dried using a freezing dryer (Ilshin Lab Co., LTD, Seoul, Korea). Dried jellyfish (36 g) fragmentized were extracted with 300 ml of 50% ethanol (EtOH) three times under reflux at 50 °C for 24 h, then filtered and concentrated to yield the EtOH extract (25 g). The EtOH extract was suspended in 100 ml $H_2O$ and extracted successively with n-hexane (Hex), ethylacetate (EtOAc; EA), and n-butanol (n-BuOH) to yield an n-hexane fraction (34 mg), an EA fraction (42 mg), an n-BuOH fraction (1.9 g), and water residue (18.4 g). The concentrated extract (34 mg) was then lyophilized, resulting in 14.9 mg of powder. Dried HE was subsequently dissolved in dimethyl sulfoxide (DMSO) diluted with DMEM media. The final concentration of DMSO was adjusted to 0.1% (v/v) in the culture media.

### Cell culture and reagents

The human CML K562 cell line, human colon cancer HCT116 cells and human liver cancer Huh-7 cells were purchased from ATCC (American Type Culture Collection; Rockville, MD, USA). The human CML K562 cell line was cultured in RPMI1640, HCT116 cells and Huh-7 cells were cultured in DMEM (WelGENE Co., Daegu, Korea) containing 10% fetal bovine serum (FBS), penicillin (100 U/mL), and streptomycin (100 mg/mL) at 5% CO2 in a humidified incubator at 37 °C. Z-VAD-FMK (a pan-caspase inhibitor)

(catalog no. 219007) was purchased from Calbiochem (Darmstadt, Germany). 3-(4,5-dimethylth-iazol-2-yl)-2,5-diphenyltetrazolium bromide (MTT) (catalog no. M2128) was purchased from Sigma–Aldrich (St. Louis, MO, USA). 6-diamidino-2-phenylindole dihydrochloride (DAPI) (catalog no. D9542) was purchased from Sigma-Aldrich (St. Louis, MO, USA). SB203580 (catalog no. 559389) and SP600125 (catalog no. 420119) were purchased from Calbiochem (Darmstadt, Germany). U0126 (catalog no. V1121) was purchased from Promega (Madison, WI, USA). Antibodies against caspase-3 (catalog no. 9661), caspase-8 (catalog no. 9746), cleaved caspase-9 (catalog no. 9501), p-JNK (catalog no. 9251), JNK (catalog no. 9252), and p-p38 (catalog no. 9211) were purchased from Cell Signaling Technology (Dancers, MA, USA). Antibodies against $\beta$-actin (catalog no. sc-47778), PARP-1 (catalog no. sc-7150), Bcl-2 (catalog no. sc-492), BAX (catalog no. sc-493), p38 (catalog no. sc-535), CDK2 (catalog no. 163), CDK4 (catalog no. sc-264), cyclin A (catalog no. sc-596), and cyclin D1 (catalog no. sc- 450) were purchased from Santa Cruz Biotechnology (Paso Robles, CA, USA). The Bio-Rad protein assay kit (catalog no. 500-0114 and 500-0113) was purchased from Bio-Rad (Richmond, CA, USA). The Annexin V-FITC/PI apoptosis detection kit (catalog no. 556547) was purchased from BD Biosciences (San Jose, CA, USA).

## MTT assay

Cell were plated in a 96-well culture plate ($5 \times 10^4$ cells/well) and treated with various concentrations (0, 10, 20, 30, 40, and 50 μg/ml) of Jellyfish-HE. After 24 h, the media was removed and MTT (0.5 mg/ml) was added to each well for 4 h. Formazan crystals from MTT reduction were dissolved in DMSO and the OD value was read at 590 nm with a Versamax microplate reader (Molecular Devices, Sunnyvale, CA, USA).

## DAPI stain assay

After treatment with Jellyfish-HE, to confirm nuclear condensation, cells were stained with DAPI. Before treatment with Jellyfish-HE, cover slides were coated with lysine to encourage attachment of K562 cells. Cells were spread in 24-well culture plates ($4 \times 10^5$ cells/well) and treated with Jellyfish-HE (40 μg/mL) for 24 h. Then, cells were washed with 1 X PBS and fixed with 4% paraformaldehyde. After 20 mins at 4 °C, the cells were washed with 1 X PBS and stained with DAPI (1 mg/mL) for 10 mins at room temperature in the dark. Then, the cells were washed with 1 X PBS and mounted with mounting solution (Dako, Glostrup, Denmark). Nuclei were detected under a fluorescence microscope TMS (Nikon, Tokyo, Japan).

## Annexin V and PI staining

After treatment with 40 μg/ml of jellyfish hexane extract for 8 h, K562 cells were harvested and cell were washed with PBS and suspended with binding buffer (1X). After that 4 μl Annexin V-FITC and 2 μl propidium iodide (PI) were added in the cells for 15 min at 37 °C in the dark. Then, cells were analyzed with a flow cytometer, FACS Canto II (BD Biosciences, San Jose, CA, USA). Using this data, Living cells (Annexin V−/PI−), early apoptotic cells (Annexin V+/PI−), late apoptotic cells (Annexin V+/ PI+), necrotic cells (Annexin V−/PI−) cells were measured.

## Western blot analysis

K562 cells were lysed in a buffer containing 50 mmol/L Tris–HCl (pH 8.0), 150 mmol/L NaCl, 100 m mol/L NaF, 100 mg/mL phenylmethylsulfonyl fluoride (PMSF), 1 mg/mL aprotinin, and 1% Triton X-100. After 30 mins at 4 °C, cells were centrifuged at 13,000 rpm at 4 °C. Then, protein concentration in the supernatant was quantified using the Bio-Rad protein assay (Bio-Rad, Berkeley, CA, USA). Equal amounts of whole cell lysates were fractionated by sodium dodecyl sulfate-polyacrylamide gel electrophoresis (SDS-PAGE). After electrophoresis, gels were transferred to polyvinylidene difluoride (PVDF) membranes using the Hoefer electrotransfer system (Amersham Biosciences, Buckinghamshire, UK). To visualize the target protein, the membranes were incubated at 4 °C overnight with each primary antibody. Then, the membranes were washed and incubated at room temperature for 1 h with the appropriate secondary antibody. Detection was carried out using a secondary horseradish peroxidase-linked antibody and the ECL chemiluminescence system (Amersham Biosciences, Buckinghamshire, UK). Films were scanned and densitometry analysis was performed using Image J software.

## Cell cycle analysis

To analyze cell cycle, K562 cells were treated with 40 μg/ml of Jellyfish-HE for 12h and cells were harvested, fixed with 70 % ethanol at 4 °C for 30 min. Fixed cells were washed with PBS and then stained PI buffer containing propidium iodide, RNase, Nacitrate and NP-40 for 30 min in the dark. After that cells were analyzed by flow cytometer, FACS Canto II (BD Biosciences, San Jose, CA, USA).

## Statistical analysis

All data are expressed as means $\pm$ SEM of three independent replicates for each group. Comparisons were made by Student's $t$-test. $^*P < 0.05$ was considered statistically significant. Statistical analysis was measured using GraphPad Prism software 5.0.

# RESULTS

## Jellyfish hexane extract specifically induces cell death in K562 cancer cells

In order to examine whether Jellyfish extracts can affect cell viability in the K562 cancer cell line, jellyfish were extracted with a variety of organic solvents (EtOH, BuOH, Ethyl Acetate, and hexane), as well as distilled water, as described in the Materials and Methods section. Then, human chronic myeloid leukemia K562 cells were treated for 3 days with 50 μg/ml EtOH, BuOH, ethyl acetate, or hexane extract. Only Jellyfish-HE obviously exhibited the reduced cell viability at the concentrations used. Cell viability was measured using the MTT assay. As shown in Fig. 1, cell viability of K562 cells was specifically decreased in Jellyfish-HE-treated cells. The results demonstrated that Jellyfish-HE specifically decreases cell viability in K562 cells, whereas the other extracts do not.

Furthermore, K562 cells have been treated with the H2O extracts, EtOH extracts, BuOH extracts, EA extracts and hexane extracts for 1 to 3 days to evaluate possible changes in acting compounds. The results have also shown that the H2O extracts, EtOH extracts,

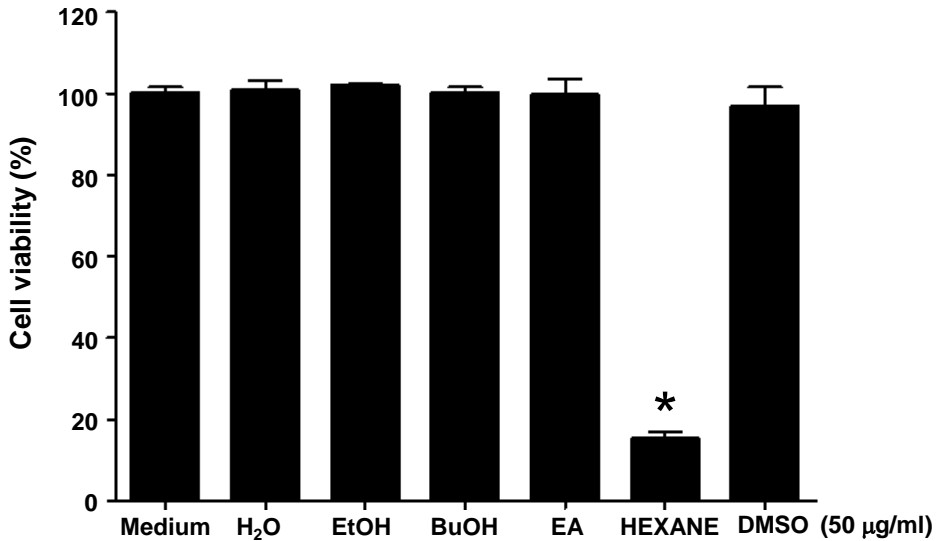

**Figure 1** **Effects of various jellyfish extracts on K562 cells.** (A) K562 cells were treated with 50 µg/ml of medium (negative control), $H_2O$ extract, EtOH extract, BuOH extract, EA extract, hexane extract, or DMSO (negative control) for three days and cell viability was measured using the MTT assay. *, $P < 0.05$, **, $P < 0.01$ and ***, $P < 0.005$ vs. control (DMSO-treated cells).

BuOH extracts and EA extracts do not affect cell viabilities, in contrast to hexane extracts in K562 cells (Fig. S1). These results indicated that the long term treatment such as 3 days does not seem to cause loss of activity of the active compounds.

## Jellyfish hexane extract induces cell death in a dose-dependent manner in various cancer cell lines

Because Jellyfish-HE inhibits the growth of human chronic myeloid leukemia K562 cells, we investigated its effects in other cancer cell lines. Several cancer cell lines, including human colon cancer HCT116 cells and human liver cancer Huh-7 cells, were treated with Jellyfish-HE and assessed for cytotoxicity. Cells were treated with various concentrations (0, 10, 20, 30, 40, and 50 µg/ml) of Jellyfish-HE for 24 h, as shown in Fig. 2. As shown in Fig. 2, the growth rate of HCT116 human colon cancer cells (Fig. 2B) and Huh-7 human liver cancer cells (Fig. 2C) were examined using an MTT assay. When half-maximal inhibition concentrations (IC50) were measured on the cancer cell lines, each IC50 value was calculated to be 49.51 µg/ml, 62.85 µg/ml and 67.28 µg/ml for K562 cells, HCT116 cells and Huh-7 cells, respectively (Fig. 2). For the calculation of IC50, the computed maximum effects on cell viability were calculated to be 33.85 % (84.23 µg/ml) in K562 cells, 34.34% (101.0 µg/ml) in HCT116 cells and 39.66 % (104.9 µg/ml) in Huh-7 cells. The results of the MTT assays showed that Jellyfish-HE inhibits cell proliferation in a dose-dependent manner in all cancer cell lines. K562 cells were especially susceptible to Jellyfish-HE-induced growth suppression, although growth of all of the tested cancer cells was inhibited.

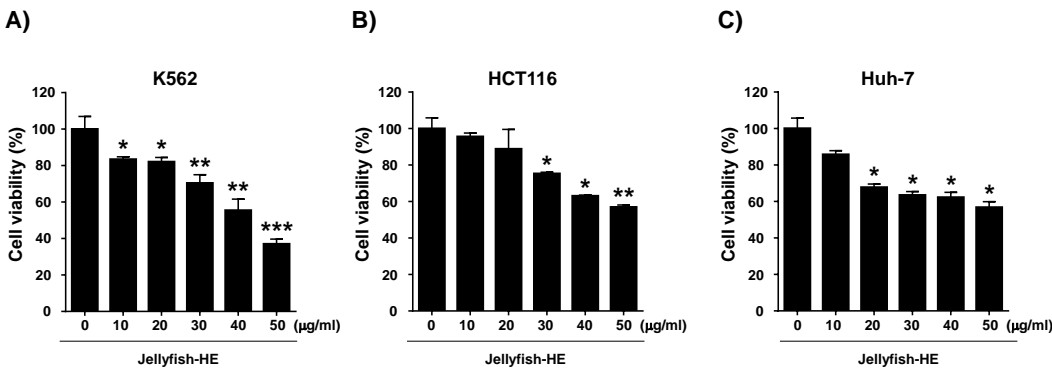

**Figure 2** **Effects of jellyfish hexane extract on various cancer cell lines, including K562 cells, human colon cancer HCT116 cells, and human liver cancer Huh-7 cells.** (A) K562, (B) HCT116, and (C) Huh-7 cell (C) lines were treated with Jellyfish hexane extract for 24 h and cell viability was measured with the MTT assay. $* P < 0.05$, $** P < 0.005$ and $*** P < 0.0005$ vs. control (untreated).

## Jellyfish hexane extract induces apoptosis characterized by typical biochemical and morphological changes, such as cell shrinkage, DNA fragmentation, chromatin condensation, and formation of apoptotic bodies in K562 cells

Apoptosis is typically characterized by cell death-related biochemical and morphological changes such as cell shrinkage, DNA fragmentation, chromatin condensation, and formation of apoptotic bodies (*Haefner, 2003*). In this study, to determine whether Jellyfish-HE induces cell death by apoptosis, K562 cells were treated with various concentrations (0, 10, 20, 30, 40, and 50 μg/ml) of Jellyfish-HE for 24 h, and then morphological changes were observed using microscopy. Apoptotic cell body-like cell morphologies were easily observed in Jellyfish-HE-treated K562 cells in a dose-dependent manner (Fig. 3A). To further examine the nuclear morphological changes in K562 cells, DAPI staining was performed after treatment with 40 μg/ml Jellyfish-HE (Fig. 3B). As shown in Fig. 3B, apoptotic body-like shapes and chromatin condensation were observed in cells treated with 40 μg/ml of Jellyfish-HE compared with untreated control cells. Next, to confirm that Jellyfish-HE-induced apoptosis is directly under the control of Jellyfish-HE, cells were treated with 40 μg/ml Jellyfish-HE for 8 h and then assessed using Annexin V-FITC/PI staining and quantitative analysis (Fig. 3C). When the statistical significance has been analyzed using Student's *t*-test, the Annexin V-positive, PI-positive and Annexin V/PI-double positive cells were significantly increased (Fig. 3D). The 3 positive levels are similar to that of the known apoptotic agents such as lactose-binding lectins (*Kwak et al., 2015*). The results clearly show that the levels of Annexin V and Annexin V-PI positive cells were increased by treatment with 40 μg/ml Jellyfish-HE. These results suggest that Jellyfish-HE induces apoptotic cell death in K562 cells.

## Jellyfish hexane extract-mediated apoptosis acts through the intrinsic and extrinsic apoptotic pathways in K562 cells

Caspases are vital in apoptosis. Therefore, after treatment of K562 cells with 40 μg/ml Jellyfish-HE, the cleaved forms of PARP and procaspase-3 through activation of caspase-8

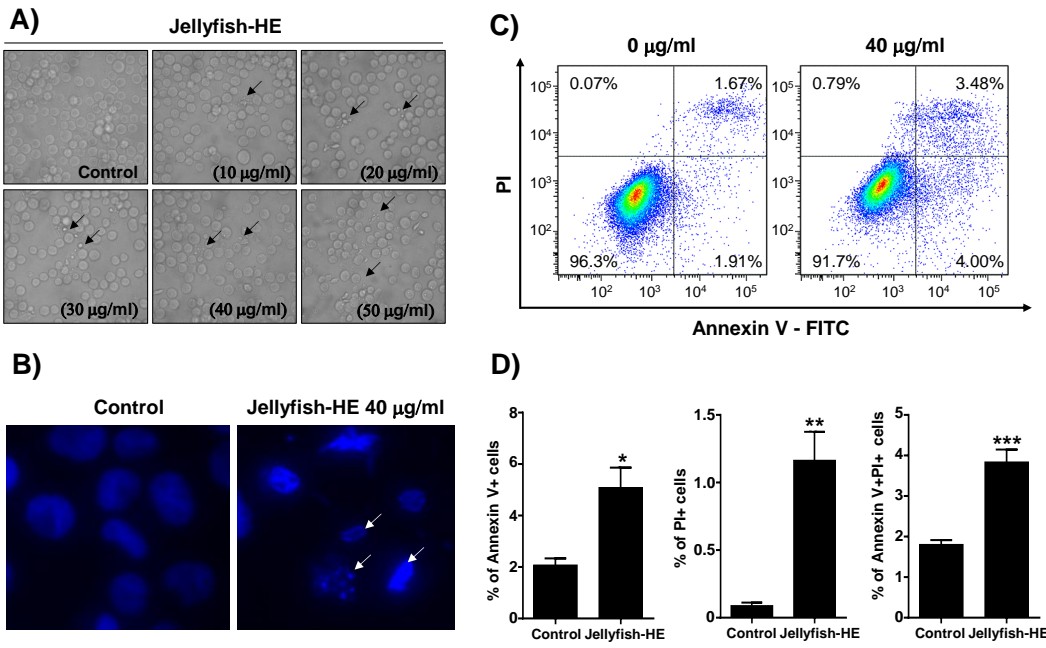

**Figure 3** **Jellyfish hexane extract induces apoptosis in K562 cells.** (A) Morphology. K562 cells were treated for 24 h with various concentrations of Jellyfish-HE. Phase contrast microscopic observation was been made using a Nikon TMS (Tokyo, Japan). Arrows indicate apoptotic bodies, which are characteristic of cell death. After 24 h incubation with or without Jellyfish-HE, nuclear fragmentation was stained by DAPI for 10 min at 37 °C by fluorescence microscopy (Zeiss Axioskop 2 microscope) (B). Arrows indicate fragmented nuclei. To observe apoptotic cell death in the earlier stages of treatment with Jellyfish-HE, cells were treated with Jellyfish-HE for 8 h. Then, apoptotic cells were detected by PI and Annexin V double staining by flow cytometry (FACS Canto II) (C) and quantitative analysis (D). * $P < 0.05$, ** $P < 0.005$ and *** $P < 0.0005$ vs. control (untreated).

and caspase-9 were investigated by immunoblotting analysis and densitometry. Cleaved PARP and mature caspase-3 levels were normalized to those of $\beta$-actin as an internal control (Figs. 4A, 4B). The results demonstrated that Jellyfish-HE induces apoptosis in a dose-dependent manner through both the extrinsic and intrinsic apoptotic pathways in K562 cells. The BCL-2 family is a group of proteins that have both pro- (BAX, BAD, and others) and anti- (BCL-2, Bcl-xL, and others) apoptotic functions. Thus, the relative ratio of those proteins expressed during the apoptotic process helps to determine the type of cell death signaling (*Gross, McDonnell & Korsmeyer, 1999*). As shown in Figs. 4C and 4D, after treatment with 40 μg/ml Jellyfish-HE for 24 h, the levels of BCL-2 and BAX were analyzed by immunoblotting and densitometry. The level of BCL-2 was decreased, while that of BAX was slightly increased and the ratio of BAX/BCL-2 was significantly increased. To further confirm that Jellyfish-HE induces apoptosis that depends on the caspase cascade, K562 cells were treated with a pan-caspase inhibitor, Z-VAD, along with 40 μg/ml Jellyfish-HE for 12 h, and cell viability was measured with the MTT assay (Fig. 4E). In the same conditions as shown in Fig. 4E, the levels of PARP and caspase-3 were also analyzed by immunoblotting (Fig. 4F). Treatment with Z-VAD and Jellyfish-HE resulted in an increase in the viability of K562 cells compared to cells incubated in the absence of Z-VAD and the

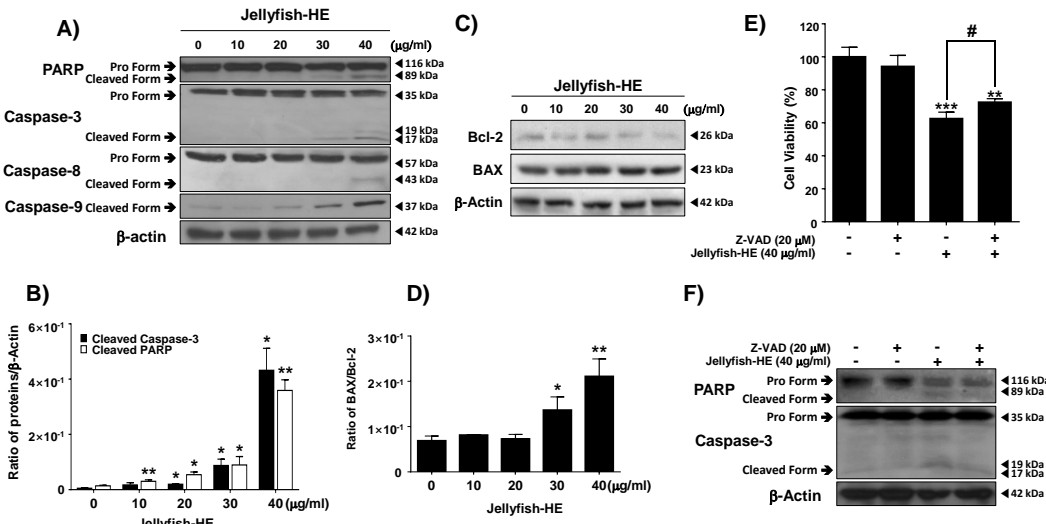

**Figure 4 Jellyfish hexane extract induces apoptosis via extrinsic and intrinsic pathways in K562 cells.**
(A) After treatment of K562 cells with various concentrations (0, 10, 20, 30, and 40 mg/ml) of Jellyfish-HE
for 24 h, the expression levels of PARP, caspase-3, caspase-8 and caspase-9 were analyzed by immunoblot-
ting with antibodies specific for PARP, caspase-3, caspase-8, and caspase-9. (B) The ratio of each protein
to $\beta$-actin was calculated using Image J software. (C) After treatment with Jellyfish-HE at various concen-
trations for 24 h, levels of Bcl-2 and BAX proteins were analyzed by immunoblotting and the BAX/Bcl-
2 ratio was quantified by densitometry (D). K562 cells were treated with 40 mg/ml Jellyfish-HE for 12 h
in the presence or absence of Z-VAD and then (E) cell viability was measured by an MTT assay and ana-
lyzed by immunoblotting (F) with antibodies specific for PARP and capase-3. b-Actin was used as a load-
ing control. * $P < 0.05$, ** $P < 0.005$ and *** $P < 0.0005$ vs. control (untreated). # $P < 0.05$ vs. treatment
with Jellyfish-HE.

presence of Jellyfish-HE. Likewise, the levels of cleaved caspase-3 and PARP were increased
in the presence of Z-VAD and Jellyfish-HE in K562 cells. Overall, these results suggest that
Jellyfish-HE induces caspase-dependent apoptosis through both the extrinsic and intrinsic
apoptotic pathways.

## Jellyfish hexane extract induces apoptosis through the p38 pathway

In caspase-regulated apoptotic cell death, the MAPK signaling pathway, which involves
ERK1/2, p38, and JNK, has been reported to play important roles, and it controls apoptosis
in human cancers (*Zhang et al., 2014*). In the present study, in order to examine whether
the MAPK signaling pathway activates Jellyfish-HE-induced apoptosis, we analyzed the
phosphorylation level of MAPK by Western blot analysis after treatment of K562 cells
with 40 $\mu$g/ml of Jellyfish-HE for various times up to 2 h and in different doses for
12 h. As shown in Figs. 5A and 5C, the levels of phosphorylated ERK, phosphorylated
JNK, and phosphorylated p38 were significantly increased in dose- and time-dependent
manners, while the total levels of such proteins remained unchanged, indicating increased
phosphorylation of the signaling molecules. To compare the relative ratio between the
phosphorylated forms and total protein forms, each was analyzed by densitometry using
Image J software (Figs. 5B, 5D). Although the three different MAPKs, ERK, JNK, and p38,
were all activated by Jellyfish-HE, a more precise role of each one may be important in the

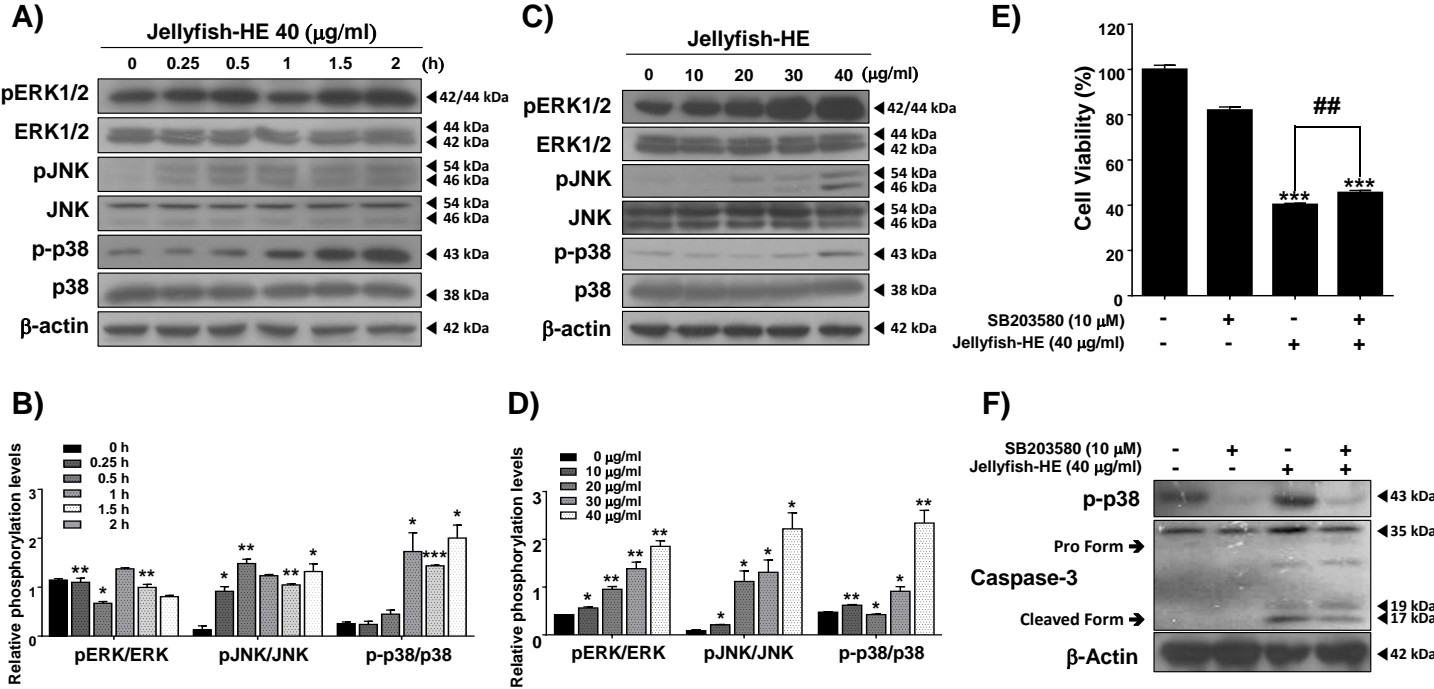

**Figure 5** **Jellyfish hexane extract induces apoptosis through the p38 MAPK pathway.** pERK1/2, ERK, pJNK, JNK, p-p38, and p38 were specifically detected by immunoblotting analysis at time intervals of 0, 0.25, 0.5, 1, 1.5, and 2 h after treatment with 40 mg/ml Jellyfish-HE (A). After treatment with various concentrations (0, 10, 20, 30, and 40 μg/ml) of Jellyfish-HE for 12 h (C), cell extracts were prepared and separated on SDS-PAGE followed by Western blot analysis. Band densities were then analyzed by densitometry using Image J software (B) and (D). After treatment of the cells with 40 μg/ml Jellyfish-HE in the presence or absence of p38 inhibitor, SB203580, cell viability was measured by the MTT assay (E) and the levels of p-p38, pro caspase-3, and cleaved caspase-3 were specifically detected using Western blots with antibodies specific to p-p38 and caspase-3 (F). β-Actin was used as an internal control. * $P < 0.05$, ** $P < 0.005$ and *** $P < 0.0005$ vs. control (untreated). ## $P < 0.005$ vs. treatment with Jellyfish-HE.

subsequent process of apoptosis. Therefore, to confirm which MAPK affects Jellyfish-HE-induced apoptosis, K562 cells were examined using specific inhibitors of MAPK signaling. Cells were pretreated for 1 h with 10 μM of specific inhibitors, U0126 (ERK inhibitor), SP600125 (JNK inhibitor), and SB203580 (p38 inhibitor), and then treated with 40 μg/ml of Jellyfish-HE for a further 12 h. The viability of the treated cells was then measured using the MTT assay. Treatment of the cells with Jellyfish-HE and SB203580 resulted in a slight increase in cell viability compared with treatment with Jellyfish-HE only (Fig. 5E). In contrast, treatment with U0126 or SP60015 failed to inhibit Jellyfish-HE-induced apoptosis (Fig. S2). To further investigate the mechanisms of apoptosis, we analyzed the expression levels of apoptotic proteins, which are specifically related to the biochemical and enzymatic processes, by Western blotting. As shown in Fig. 5F, the level of cleaved caspase-3 (detected at 17 kD and 19 kD), a well-known indicator of apoptosis, is decreased when 10 μM of SB203580 was applied for 1 h followed by 40 μg/ml of Jellyfish-HE treatment for 12 h. These results indicated that p38 MAPK signaling is directly related to Jellyfish-HE-induced apoptosis in K562 cells.

### Jellyfish hexane extract induces cell cycle arrest

As shown in Fig. 1, Jellyfish-HE decreased the growth rate of K562 cells. Therefore, we investigated whether the Jellyfish-HE extract affects cell cycle. K562 cells were treated with 40 µg/ml of Jellyfish-HE or absence of Jellyfish-HE for 12h. Jellyfish-HE treated K562 cells were arrested in G0/G1phase (Figs. 6A and 6B). To further investigate, whether the cell cycle-related proteins CDK2, CDK4, cyclin A, and cyclin D1 are altered in Jellyfish-HE-treated cells. K562 cells were treated with various concentrations (0, 10, 20, 30, and 40 µg/ml) of Jellyfish-HE for 24 h and cell extracts were subjected to immunoblotting and densitometry analysis. Cyclin D1 and CDK4 are known to be expressed in the initial stages of the G1 and S1 phase transitions (Liu et al., 2014). Therefore, downregulation of Cyclin D1 and CDK4 is regarded as an indicator of G0/G1 cell cycle arrest. In this study, as shown in Fig. 6C, treatment of the cells with various concentrations of Jellyfish-HE significantly decreased the expression levels of the G0/G1 cell cycle-related proteins Cyclin D1 and CDK4 in a dose-dependent manner. Other proteins involved in the cell cycle, CDK2 and cyclin A, were also downregulated by Jellyfish-HE treatment (Fig. 6D). Although the expression levels of CDK2, CDK4, and Cyclin D1 proteins were up-regulated by HE extract treatment at low dose (10 µg/ml), the adaptation state of the cells is suggested. In general, most cells respond to environmental changes such as stress, which called "Adaptation" for cell survival, as previously reported (Brooks et al., 2011). However, high doses of Jellyfish-HE directly induce cell injury, cell death and cell cycle arrest. These results show that Jellyfish-HE induces G0/G1 cell cycle arrest that is regulated by cell cycle-related proteins in K562 cells.

## DISCUSSION

Jellyfish is well known as a marine animal with venomous tentacles. However, the detailed compounds and structure-based molecular targets in hosts responsible for the toxic effects of jellyfish venom are unknown (Li et al., 2014). Recently, jellyfish have become a worldwide ecological problem as their population is growing rapidly. An innovative system for jellyfish disposal has not been developed. Recently, several pharmacological activities have been reported from the Jellyfish (Kang et al., 2009; Ovchinnikova et al., 2006; Yu et al., 2005) and thus, we have investigated the anti-cancer activities of jellyfish extracts against human chronic myeloid leukemic cells. In our investigation into CML-specific drugs, we have found that jellyfish hexane extract induces cell death through apoptosis and cell cycle arrest in K562 CML cells. The mechanism of action of jellyfish hexane extract, induction of apoptosis, was studied with pharmaceutical applications in mind.

In the present study, with an eye towards developing alternative treatments using natural products (Lucas et al., 2010), we investigated the anti-cancer effect of Jellyfish-HE on apoptosis and cell cycle arrest. Apoptosis is a mode of programmed cell death that regulates cellular function for organismal homeostasis, survival, and cell death (Elmore, 2007). After treatment with Jellyfish-HE, apoptotic cell bodies (Fig. 2A) and nuclear condensation, which are characteristic of apoptosis, were observed in K562 cells. Furthermore, caspases are important proteins for controlling cell death and inflammation. In this study, caspase-3

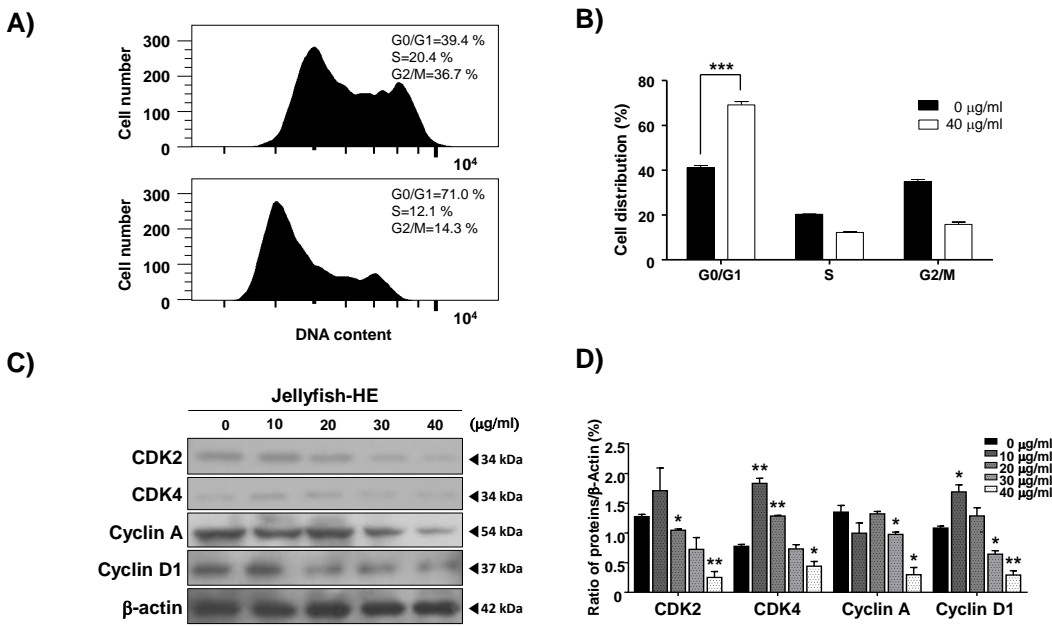

**Figure 6** **Jellyfish hexane extract induces cell cycle arrest.** (A) After treatment of cells with or without 40 µg/ml Jellyfish-HE, DNA contents were analyzed by flow cytometer, FACS Canto II. (B) Analysis of cell distribution was quantified using Graph Pad Prism software 5.0. (C) After treatment of cells with various concentrations (0, 10, 20, 30, and 40 µg/ml) of Jellyfish-HE for 24 h, the cells were analyzed by immunoblotting with antibodies for CDK2, CDK4, Cyclin A, and cyclin D1. $\beta$-Actin was used as a loading control. (D) Band intensities were quantified using Image J software. $* P < 0.05$ and $** P < 0.005$ vs. control (untreated).

and PARP were detected as markers of apoptosis (Fig. 3B). Caspase-3 and PARP are cleaved by activated apoptotic caspases (*Ouyang et al., 2012*). When K562 cells were treated with Jellyfish-HE, we observed activation of caspase-3 and PARP in a dose-dependent manner. Moreover, when a pan-caspase inhibitor, Z-VAD, was applied before Jellyfish-HE, we observed an increase in cell viability as well as inhibition of cleaved PARP and caspase-3. Cleavage of caspase-3 and PARP are well known as classic apoptosis markers in many cancer cells (*Motomura et al., 2008*; *Ola, Nawaz & Ahsan, 2011*), and two pathways, the extrinsic and the intrinsic pathway, are involved. The extrinsic apoptotic pathway, which is caused by DRs, is regulated by caspase-8, while intrinsic apoptosis, which involves the mitochondria, is associated with activated caspase-9 and activated caspase-3, which is cleaved by caspase-8 and caspase-9 (*McIlwain, Berger & Mak, 2013*). Treatment with Jellyfish-HE induced activation of caspase-8 and caspase-9 (Figs. 3A and 3B), implying that Jellyfish-HE induces apoptosis via both the intrinsic and extrinsic apoptotic pathways. However, how Jellyfish-HE induces the extrinsic apoptotic pathway is unclear. Interestingly, our results showed decreased Bcl-2 and increased BAX protein expression (Fig. 3C). Based on the BAX/Bcl-2 ratio, we demonstrated that Jellyfish-HE induces the intrinsic apoptosis pathway in the K562 cell line (Fig. 3D). Thus, the intrinsic apoptosis pathway, mediated by the Bcl-2 family, including Bcl-2 and BAX, is affected by Jellyfish-HE. Bcl-2 was initially discovered in B cell lymphoma. The Bcl-2 family regulates both anti-apoptotic

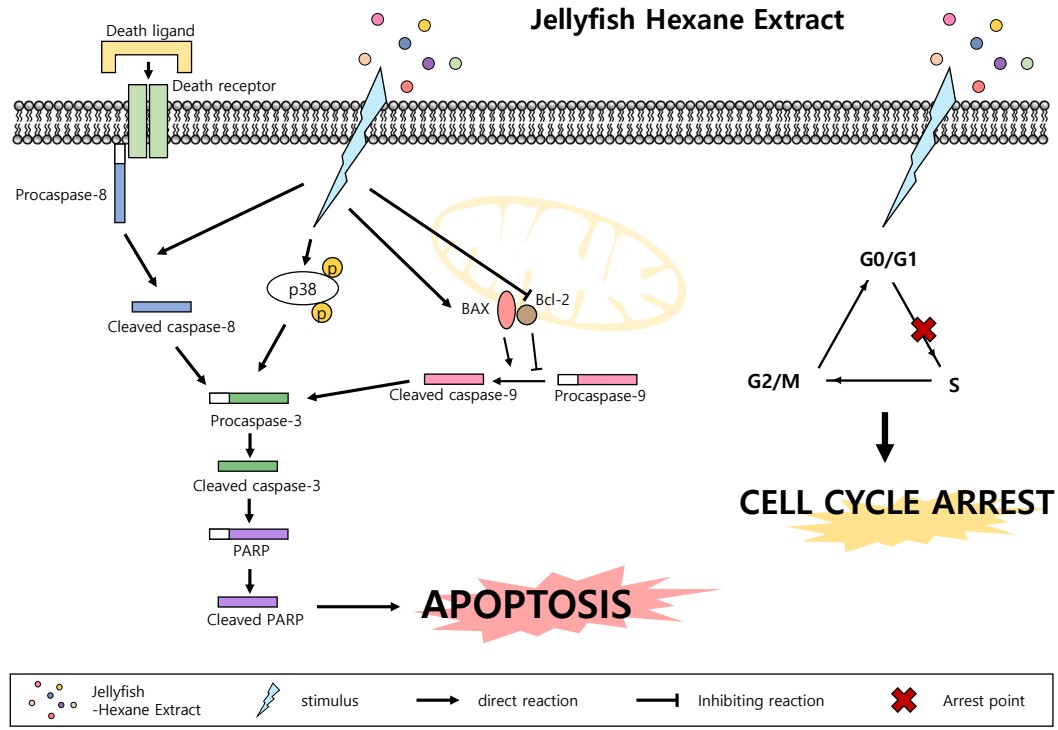

**Figure 7** **Jellyfish hexane extract induces apoptosis through the p38 pathway and cell cycle arrest.** Note that small colorful dots, blue lighting, black arrows and black hammer-head line represent the Jellyfish hexane extract, Jellyfish hexane extract-induced stimulus, direct reactions and each inhibiting reaction, respectively.

and pro-apoptotic proteins and is correlated with the activity of mitochondria. Bcl-2 is an anti-apoptotic protein that regulates calcium homeostasis. BAX, on the other hand, is a well-known pro-apoptotic protein that stimulates release of cytochrome C (*Danial, 2007*).

MAPK family members also have important role in apoptosis. MAPKs include ERKs, c-JNKs, and p38 kinase. Typically, JNK and p38 kinase are known as cell death signals, whereas ERK is a survival signal (*Osaki & Gama, 2013*). However, it has been demonstrated that ERK is involved in both cell survival and cell death in some conditions. Namely, phosphorylated ERK has been associated with apoptosis (*Cagnol & Chambard, 2010*). In the present paper, after treatment with Jellyfish-HE, cells demonstrated an upregulation in phosphorylated ERK, JNK, and p38 kinase in time- and dose-dependent manners. However, treatment with specific MAPK inhibitors along with Jellyfish-HE in K562 cells did not block apoptosis, except for an inhibitor of p38 kinase (Figs. 5A and 5B). Thus, we suggest that p38 kinase is specifically related to Jellyfish-HE-induced apoptosis in K562 cells (Figs. 5E and 5F). Our results also clearly showed that Jellyfish-HE alters cell cycle-related proteins; it induces decreases in CDK2, CDK4, Cyclin A and Cyclin D1 levels in K562 cells (Figs. 6C and 6D). Thus, we suggest that Jellyfish-HE induces cell cycle arrest by reducing the levels of CDK2, CDK4, Cyclin A, and Cyclin D1. However its exact mechanism of action is not clear. To date, several approaches have been tried to treat the patients even without complete efficacy. For the successful cases, several

CML-therapeutic drugs including imatinib (known as Glivec) (*Henkes, Van der Kuip & Aulitzky, 2008*), dasatinib (*Hochhaus & Kantarjian, 2013*) and nilotinib (*Saglio et al., 2010*) have been clinically treated. However, such effective drugs are recently suffered from the occurrence of the unknown drug resistances in the patients (*Gómez-Almaguer et al., 2016*). Therefore, chemotherapeutic drugs against CML have been subjected to search from the natural resources (*Kim, 2016*). In conclusion, Jellyfish-HE is a potential novel cancer therapy for CML which induces apoptosis and cell cycle arrest. As shown in Fig. 7, although jellyfish is a stinging organism and its known toxic compounds are reported to be water-soluble polypeptides, the present results clearly suggest that Jellyfish-HE contains valuable compounds that can target cancer cells, such as malignant leukemic cells.

### Funding

This work was supported in part by the National Research Foundation of Korea (NRF) grant funded by the Korean government (MSIP) (No. NRF-2014R1A4A1071040 to HW Chang) and (No. NRF-2015R1D1A1A01057153 to CH Kim). The funders had no role in study design, data collection and analysis, decision to publish, or preparation of the manuscript.

### Grant Disclosures

The following grant information was disclosed by the authors:
National Research Foundation of Korea (NRF): NRF-2014R1A4A1071040, NRF-2015R1D1A1A01057153.

### Competing Interests

Cheorl-Ho Kim is an Academic Editor for PeerJ.

### Author Contributions

- Sun-Hyung Ha conceived and designed the experiments, performed the experiments, wrote the paper, prepared figures and/or tables, reviewed drafts of the paper.
- Fansi Jin conceived and designed the experiments, performed the experiments.
- Choong-Hwan Kwak, Fukushi Abekura, Jun-Young Park, Tae-Wook Chung, Ki-Tae Ha and Jong-Keun Son analyzed the data.
- Nam Gyu Park analyzed the data, contributed reagents/materials/analysis tools, reviewed drafts of the paper.
- Young-Chae Chang contributed reagents/materials/analysis tools.
- Young-Choon Lee analyzed the data, contributed reagents/materials/analysis tools, source supply.
- Hyeun Wook Chang and Cheorl-Ho Kim conceived and designed the experiments, wrote the paper, reviewed drafts of the paper.

### Data Availability

The raw data has been supplied as Supplementary Files.

## Supplemental Information

Supplemental information for this article can be found online at http://dx.doi.org/10.7717/peerj.2895#supplemental-information.

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
