# Peer review of "Jellyfish extract induces apoptotic cell death through the p38 pathway and cell cycle arrest in chronic myelogenous leukemia K562 cells"

_PeerJ, doi:10.7717/peerj.2895_

## Round 0.1 · original submission · Major Revisions

All reviewers agree that your manuscript is interesting and should be published. They do point out that the use of a single malignant cell line and no healthy cell lines makes this study somewhat less compelling, both in terms of applicability to the chemotherapy of other cancers and to the toxicity towards healthy cells. Since PeerJ does not review on perceived impact but rather on technical soundness, I will not require you to include additional experiments on additional cell lines but I will need you to clearly state the limitations brought about by the lack of those studies.

Please ensure that the discussion does not duplicate information present in the Introduction., and include the other data requested by reviewers (IC50, technical details, whether your observations explain general toxicity of jellyfish extract or additional long-term effects unrelated to the immediate painful sensations ellicited by the animal's sting, etc.).

-- Additional review-level observations by the editor ---

page 4 "For that reason, natural compounds are increasingly considered important treatments that have fewer side effects than do chemical compounds. " In principle, there are no reasons why a "natural compund" would lead to fewer side effects (vs. a "non-natural compound") when delivered to an unrelated cell/organism. Please rephrase.

page 5 "In this study, we carried out activity-based pharmacological assays using jellyfish extracts" Since you have examined the extract from a single jellyfish species, it might be more accurate to rephrase this as ""In this study, we carried out activity-based pharmacological assays using extract from Nomura's jellyfish"

page 5 (materials and methods) Please include more detail on the extractions. Specifically A) how jellyfish was dried prior to ethanol extraction. B) amount of ethanol used in each extraction c) whether each ethanol extraction took 24 h under 50ºC reflux or whther that is the total time taken be three successive extractions. D) volume of organic solvents for the successive extractions from the EtOH dissolved in 100 mL H2O.

legend to figure 3: please state instrumental/technical details for panel B.

fig.3 Panel C : Axis should include numeric scale
page 9: "the results clearly show that the levels of Annexin V and Annexin V-PI positive cells were increased by treatment with 40 ug/ml Jellyfish-HE." It is hard for the non-specialist reader to evaluate how the magnitude of Annexin V-PI positive ratio compares to other pro-apoptotic compounds. Comparison with other compounds (e.g. https://doi.org/10.7717/peerj.1476) suggests that the activity of the jellyfish-HE is quite moderate. Please compare the efficacy of jellyfish-HE with other literature reports explicitly.

fig. 4, panels A/B . The ratios in the graph (B) seem to imply that the faint bands of the cleaved forms of PARP and caspase-3 actually amount to as much (or more) protein as the very strong bands of the beta-actin. How can this be? Should the y-axis go from 0 to 6*10-2 or from 0 to 6*10-1 instead of the 0-6 range shown?

fig. 4, panels C/D . The ratios in the graph (D) also seem to imply that the faint bands of Bcl-2 contain more protein than the very strong bands of the BAX.

page 11 " In contrast, treatment with U0126 or SP60015 failed to inhibit Jellyfish-HE-induced apoptosis (data not shown)." Please show this data, even if only as Supporting Information.

fig. 5 panel B: the legend states "1.2 h" instead of the intended "2 h"

ref 19 is not "Cent Nerv Syst Agents Med Chem., in press" but "Cent Nerv Syst Agents Med Chem. 15:68-73"

Reviewer 1 ·

Basic reporting

Introduction includes unecessary information about jellyfish and other well known cellular process such as apoptosis. If there is, literature should be reviewed about the effect of any kind of jellyfish extract on cancer.

Discussion seems the replication of introduction. Unneeded information about apoptosis, cell cycle etc. Literature should be discussed in combination with their data. As I check the literature, there are a few articles about jellyfish species and different cancers. They r not discussed.

Experimental design

Annexin V/PI staining procedure should be explained in detail.
Cell cycle analysis should be performed by flow cytometry to see which phases are affected. Then, related CDKs and cyclines can be checked to confirm and solve the mechanism. Statistical analysis should be performed by more sophisticated programmes such as graph pad.
Health control cells should be used to confirm cancer specific effects of extract.
What is the IC50 value of the extract?

Validity of the findings

Results are meaningfull and support their hypothesis. However, p38 pathway should be checked using down regulation and over expression of related genes.

Reviewer 2 ·

Basic reporting

No Comments

Experimental design

No Comments

Validity of the findings

No Comments

Additional comments

The authors investigate how jellyfish extract induces apoptotic cell death through the p38 pathway and cell cycle arrest in a cell culture system. It is a fairly comprehensive investigation into the signal pathways involved.
Some specific notes:
1. In general, the identification of the component in the extract that elicits the said effects is absent, the study is largely descriptive.
2. Only cancer cell lines were used to evaluate the cytological effects of hexane extract, but a normal cell line control was not included.
3. Is the cell toxicity of jellyfish extract related to any of the events measured in this study?

Reviewer 3 ·

Basic reporting

Kim and colleagues have investigated the anti-cancer activity of hexane extracts from Nomura's jellyfish using chronic myelogenous leukemia cell line, K562 cells as a model. Jellyfish hexane extract induced apoptotic cell death in K56 cell line. Jellyfish-HE markedly arrests cell cycle in G0/G1 phase. ThJellyfish-HE mediated apoptosis was found to be mediated via activation of caspase-3,-8 and -9. In addition, apoptosis was correlated with phosphorylation of p38, JNK and ERK1/2. Furthermore, Jellyfish-HE-induced apoptosis was blocked by a generalized caspase inhibitor, Z-VAD. Interestingly phosphorylation of p38 MAPK was inhibited by pretreatment of SB203580, inhibitor of p38. SB203580 treatment of K562 also blocked jellyfish-HE-induced apoptosis. Additionally, Jellyfish-HE markedly arrests cell cycle in G0/G1 phase.

In summary it is an interesting study showing anticancer effects of Jellyfish-HE in CML .

Major issue: Only one cell line has been use that limit to generalized the anticancer effects. Author should show at least three cell lines.

Minor: The manuscript needs thorough editing as it has several spelling and grammar mistakes.

Experimental design

ok

Validity of the findings

needs more cell line to validate the conclusion

---

## Round 0.2 · Major Revisions

I need to request a correction before I send this revised version back to the original reviewers. The issues are the following:

- in p.9, authors state "When half-maximal inhibition concentrations (IC50) were measured on the cancer cell lines, each IC50 value was calculated to be 49.51 mg/ml, 31.53 ug/ml and 13.18 ug/ml for K562 cells, HCT116 cells and Huh-7 cells, respectively (Figure 2)." In Figure 2, however, inhibition of cell growth in cell lines HCT116 and Huh-7 does not reach 50%, even at concentration of 50 ug/mL. It is not at all clear how your numbers were computed and how they could possibly be correct. Besides, the experimental error bars are not compatible with a precision of 4 significant figures.

- The original supplier of the HCT116 cells and Huh-7 cells used should also be disclosed.

---

## Round 0.3 · Minor Revisions

You have addressed the reviewers' queries in a most satisfactory manner. Please address the small changes requested by the reviewer. For the IC50 computations, please include also the computed maximum effect of each compound.

Suggested language edits:

"Furthermore, K562 cells have been treated with the H2O extracts, EtOH extracts, BuOH extracts, EA extracts and hexane extracts for 1 to 3 days to prevent the possible changes in acting compounds. The results have also shown that the H2O extracts, EtOH extracts, BuOH extracts and EA extracts do not affect any changes in the cell viabilities, except for hexane extracts in K562 cells (Supplementary Figure 1). These results indicated that the long term treatment such as 3 days does not affect changes, including degradation, in acting compounds."

might be more idiomatic if written as

"Furthermore, K562 cells have been treated with the H2O extracts, EtOH extracts, BuOH extracts, EA extracts and hexane extracts for 1 to 3 days to evaluate possible changes in acting compounds. The results have also shown that the H2O extracts, EtOH extracts, BuOH extracts and EA extracts do not affect cell viabilities, in contrast to hexane extracts in K562 cells (Supplementary Figure 1). These results indicated that the long term treatment such as 3 days does not seem to cause loss of activity of the active compounds."

Reviewer 2 ·

Basic reporting

No Comments

Experimental design

No Comments

Validity of the findings

No Comments

Additional comments

This revised manuscript was obviously improved. Therefore, I recommend the publication of the present paper as is.
One specific note:
1. Materials and Methods, line 105-106, “the harvested jellyfish (100 g) was vacuum-dried using a freezing dryer (Ilshin Lab Co., LTD, Seoul, Korea). Dried jellyfish (36 g) fragmentized were extracted with……”. Please make sure that how many dried products (36 g ?) were got from 100 g of the harvested jellyfish, considering their high water content.

---

## Round 0.4 · Minor Revisions

Thank you for your prompt reply to the last remaining reviewer requests. For all purposes, please consider this decision an "Accept". I am afraid, though, that I did not adequately phrase my request for "computed maximum effect of each compound": besides the concentration required for that effect, I think the magnitude of the maximum effect (in % cell viability) should also be included.

---

## Round 0.5 · accepted · Accept

Thank you for addressing all remaining issues.